# The Effects of Differences in the Morphologies of the Ulnar Collateral Ligament and Common Tendon of the Flexor-Pronator Muscles on Elbow Valgus Braking Function: A Simulation Study

**DOI:** 10.3390/ijerph18041986

**Published:** 2021-02-18

**Authors:** Masahiro Ikezu, Mutsuaki Edama, Takuma Inai, Kanta Matsuzawa, Fumiya Kaneko, Ryo Hirabayashi, Ikuo Kageyama

**Affiliations:** 1Institute for Human Movement and Medical Sciences, Niigata University of Health and Welfare, Niigata 950-3198, Japan; edama@nuhw.ac.jp (M.E.); hwd17001@nuhw.ac.jp (T.I.); hpm19009@nuhw.ac.jp (K.M.); hpm18006@nuhw.ac.jp (F.K.); hirabayashi@nuhw.ac.jp (R.H.); 2Department of Anatomy, School of Life Dentistry at Niigata, Nippon Dental University, Niigata 951-8151, Japan; kageyama@ngt.ndu.ac.jp

**Keywords:** elbow, anatomy, ulnar collateral ligament injury, elbow valgus restriction, three-dimensional model, strain, simulation, baseball

## Abstract

The anterior bundle (AB) and posterior bundle (PB) of the ulnar collateral ligament and the anterior common tendon (ACT) and posterior common tendon (PCT) of the flexor-pronator muscles have an independent form and an unclear form. The purpose of this study was to clarify the effect of differences in the morphologies of the AB, PB, ACT, and PCT on the elbow valgus braking function. This investigation examined three elbows. In the classification method, the AB, PB, ACT, and PCT with independent forms constituted Group I; the AB, ACT, and PCT with independent forms and the PB with an unclear form constituted Group II; the AB, PB, ACT, and PCT with unclear forms constituted Group III. The strains were calculated by simulation during elbow flexion at valgus at 0° and 10°. At 0° valgus, Group I and Group II showed similar AB and PCT strain patterns, but Group III was different. At 10° valgus, most ligaments and tendons were taut with increasing valgus angle. The average strain patterns of all ligaments and tendons were similar for the groups. The AB, PB, ACT, and PCT may cooperate with each other to contribute to valgus braking.

## 1. Introduction

Ulnar collateral ligament (UCL) injury is among the frequent sports injuries of throwing athletes [1]. The mechanism of UCL injury is thought to involve repeated valgus stress on the elbow during throwing motions [2]. The UCL is composed of the anterior bundle (AB), the posterior bundle (PB), and the transverse bundle. Of these, the AB is the primary restraint against elbow valgus stress, and the PB is the secondary restraint [3]. Furthermore, the AB is functionally divisible into anterior, central, and posterior bands, and these three bands have been reported to show different strain patterns during elbow flexion and extension [4,5,6,7,8].

Ciccotti et al. [5] measured the strain of the anterior, central, and posterior bands of the AB in fresh cadavers. They reported that the anterior band is isometric through the elbow flexion range, and that the central and posterior bands are taut with elbow flexion. However, Cohen et al. [6] measured the strain of the AB in fresh cadavers using the same methods as Ciccotti et al. [5], and they reported that the anterior band is taut with elbow extension, the central band is isometric through the elbow flexion range, and the posterior band is taut with elbow flexion. On the other hand, the PB has been reported to be taut with elbow flexion [4,8], and it is taut not just with elbow flexion, but also with elbow extension [4]. Thus, the reported AB and PB strain patterns during elbow flexion and extension are inconsistent. This may be related to differences in anatomical features.

Regarding anatomical features, it has been reported that the AB forms were cord-shaped and fan-shaped [9]. The PB is a thickened posterior joint capsule [10], and there is no clear definition. In recent years, it has been reported that the AB and common tendon (CT) of the flexor-pronator muscles can be separated macroscopically [11], while others have stated that they cannot be separated [12]. Thus, the reported anatomical features of the AB, PB, and CT have been controversial. Therefore, in previous research [13,14] the anatomical features of the AB, PB, and CT were examined, and the AB, PB, and CT were found to have an independent form and unclear form. Other ligaments have been reported to have a different braking function due to differences in anatomical features [15,16]. However, no one has examined the effects of differences in the morphologies of the AB, PB, and CT on the elbow valgus braking function.

Therefore, the purpose of this study was to construct three-dimensional models of the AB, PB, and CT and to clarify the effects of differences in the morphologies of the AB, PB, and CT on elbow valgus braking function by simulation.

## 2. Materials and Methods

### 2.1. Cadavers

This investigation examined three elbows from three Japanese cadavers (mean age at death, 86 ± 3 years; two sides from men and one side from a woman; all three were left sides) that had been switched to alcohol after placement in 10% formalin. No sides showed signs of previous major surgery around the upper extremity.

### 2.2. Methods

In the dissection procedure, the skin, subcutaneous tissue, and muscular parts were removed, and the AB, PB, anterior common tendon (ACT), and posterior common tendon (PCT) were carefully dissected. Classification of the AB, PB, ACT, and PCT was performed with reference to previous studies [13,14]. Group I was classified as follows: the AB and PB could be separated as a single fiber bundle, and the AB, ACT, and PCT could be separated from each other. Group II was classified as follows: the AB, ACT, and PCT could be separated from each other, and the PB continued into the joint capsule and had an unclear form. Group III was classified as follows: the AB, PB, ACT, and PCT continued into each other, and each structure had an unclear form. 

All measurements were performed with elbow flexion at 90° and neutral rotation with reference to previous studies (Figure 1) [5,6]. 

The specimen was fixed to the examination table using nails. A single specimen of each group was then digitized using the MicroScribe system (G2XSYS, Revware, Raleigh, NC, USA) (Figure 1 and Figure 2). 

The ACT and PCT were digitized at a total of two points at the proximal and distal ends of the attachment and defined as the ACT and PCT, respectively. The AB was digitized at a total of six points of the origin and insertion, and they were defined as the anterior band, central band, and posterior band, respectively. The PB was digitized at a total of eight points of the origin and insertion, and they were defined as the anterior band, anterior central band, posterior central band, and posterior band. In previous research [13,14], the PB of Group I originated from the medial epicondyle of the humerus, and the average width of the origin was about 4 mm. Thus, the digitized parts of the anterior central band and posterior central band in Group II and Group III were the fiber bundles or joint capsules located about 2 mm anterior and 2 mm posterior from the medial epicondyle tips of the humerus, respectively. Rhinoceros 3D software (McNeel, Seattle, WA, USA) was used to construct the three-dimensional models (Figure 3).

The elbow flexion/extension axis and the varus/valgus axis were defined according to a previous study [17]. The Y-axis was defined as the line connecting the glenohumeral rotation center and the midpoint between the medial epicondyle tips of the humerus and the lateral epicondyle tips of the humerus. The glenohumeral rotation center was calculated using least-squares by digitizing about 40 data points on the humeral head [18]. The X-axis was defined as the line perpendicular to the plane formed by the medial epicondyle tips of the humerus, the lateral epicondyle tips of the humerus, and the glenohumeral rotation center. The Z-axis was defined as the common line perpendicular to the Y-axis and X-axis. Rotation around the X-axis and Z-axis indicates varus/valgus and flexion/extension, respectively. The simulations were used to calculate AB, PB, ACT, and PCT strain (%) during elbow flexion at 0°, 30°, 60°, 90°, and 120° and elbow valgus at 0° and 10°. In previous research [19], the UCL ultimate angle was about 16° with respect to elbow valgus torque. Therefore, the elbow valgus angle was set to 10° in the present study. Using the following formula, AB, PB, ACT, and PCT strains were expressed as the percentage of change of ligament and tendon length from the initial limb position (LTS), with elbow flexion at 90° and elbow valgus at 0°, to the final position (LT) after motion [20]:(1)$$\mathrm{Strain}\left(\%\right)=\left[\left(\frac{L^{T}-L_{S}^{T}}{L_{S}^{T}}\right)\times 100\right]$$

SCILAB code version 6.0.2 (Enterprises, Orsay, France) was used for the analysis.

The MicroScribe system is an instrument with high precision (measurement precision of 0.23 mm according to the manufacturer’s specifications). However, measurements must be performed manually. In addition, although the study cadavers were thoroughly fixed to the examination table such that they did not move, it was necessary to test whether they had moved, since the measurements entailed dissection of the ligament and tendon tissue. A previous study found the intraclass correlation coefficient (1, 1) to be 0.97–0.99 [21], which indicates a high level of reliability and reproducibility.

## 3. Results

### 3.1. Changes in Strain for Each Group during Flexion and Extension 

The ACT was taut with elbow extension in all groups. The PCT was taut with elbow flexion in Group I and Group II, but the PCT in Group III was shortened with elbow flexion. For the AB, the anterior band was taut with elbow extension in Group I and Group II, and the posterior band was taut with elbow flexion in Group I and Group II. However, all bands of the AB were taut with elbow extension in Group III. The PB strain patterns were different for the groups (see Figure 4 and Figure 5, Table 1).

### 3.2. Changes in Strain When Elbow Valgus (10°) Was Added at Elbow Joint Flexion and Extension in Each Group 

The posterior band of the PB in Group I was shortened at the elbow flexion position. In addition, the PCT and posterior band of the PB in Group II, as well as the posterior central and posterior bands of the PB in Group III were shortened at the elbow extension position. The other ligaments and tendons were taut with increasing elbow valgus angle in all groups. The average strain, i.e., the average of the strains of all ligaments and tendons, showed a similar strain pattern for the groups (see Figure 6, Table 2).

## 4. Discussion

This study clarified the effects of differences in morphologies of the AB, PB, and CT on elbow valgus braking function. To the best of our knowledge, no research focusing on differences in the morphologies of the AB, PB, and CT and elbow valgus braking function has been previously reported.

At 0° elbow valgus, Group I and Group II showed similar PCT and AB strain patterns, but Group III was different. The PB strain patterns were different for the groups. In previous research, the anterior and central bands of the AB were reported to be taut with elbow flexion and extension [4,5,6,8] and isometric through the elbow flexion range [5,6,7]. On the other hand, the PB strain pattern has also varied between reports [4,8]. An anatomical study of the medial elbow reported that the ACT was attached just anterior to the AB, and the PCT was attached just posterior to the AB [11]. Furthermore, it has also been reported that the traditional AB might have measured the CT as the AB [12]. In the present study, the AB, PB, ACT, and PCT were clearly defined and constructed as three-dimensional models. Therefore, the results of this study appear highly accurate. In addition, in previous research [13,14], the anatomical features of the AB, PB, and CT were examined by focusing on positional relationships with surrounding structures, and it was reported that the AB, PB, and CT were each classified into an independent form and an unclear form. In other simulation studies [15,16], the lateral collateral ligament of the ankle was classified based on morphological differences, and the strain during ankle movements has been reported to differ for each type. Therefore, the differences in the morphologies of the AB, PB, and CT might have different effects on strain during elbow flexion and extension.

At 10° elbow valgus, most ligaments and tendons were taut with increasing elbow valgus angle. In addition, the average strain showed a similar strain pattern for the groups. The AB has been reported to be the primary restraint, and the PB the secondary restraint [3]. Furthermore, in the previous studies in cadavers and in vivo, the flexor-pronator muscles including the pronator teres (PT), flexor carpi radialis, palmaris longus, flexor digitorum superficial (FDS), and flexor carpi ulnaris (FCU) muscles were reported to contribute to valgus stabilization of the elbow [22,23,24,25,26,27,28,29,30,31,32,33]. In addition, Hoshika et al. [12] reported that the tendinous septa between the PT and FDS, the tendinous septa between the FDS and FCU, the medial part of the brachialis tendon, and the deep FDS and FCU aponeuroses formed a tendinous complex. Therefore, it has been suggested that the PT, FDS, FCU, and brachialis muscles do not work independently, but that they work together and transmit muscular power to the humeroulnar joint via the tendinous complex. The average strain pattern was similar for the groups in the present study; thus, the AB, PB, ACT, and PCT may cooperate with each other to contribute to elbow valgus braking. In the future, research focusing on the relationship between the flexor-pronator muscles, which are thought to function as dynamic stabilizers, and the elbow valgus braking function will be needed.

The limitation of this study was that it involved simulations with cadavers. Therefore, mechanical properties [34,35], muscle activity, gravity, and specimen-specific effects were not considered. In addition, the number of specimens in each group was only one. Furthermore, the joint axis during elbow flexion and extension in vivo was not considered [36,37,38]. However, the purpose of this study was to compare elbow valgus braking functions among the groups. The definition of the joint axis in this study was unified among the groups. Therefore, it was thought that one could compare the elbow valgus braking functions due to differences in the form of each structure.

## 5. Conclusions

The strain patterns of the AB, PB, and PCT were different for the groups. However, the average strain pattern at 10° elbow valgus was similar for the groups. Therefore, the AB, PB, ACT, and PCT might cooperate with each other to contribute to elbow valgus braking. In the future, ulnar collateral ligament injuries in throwing athletes will need to be studied considering AB, PB, ACT, and PCT as a complex.

## Figures and Tables

**Figure 1 ijerph-18-01986-f001:**
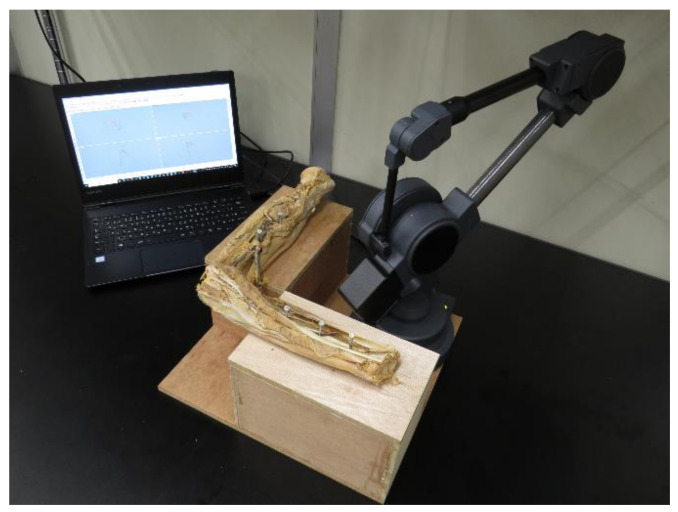
The specimen fixed on the examination table and the MicroScribe system.

**Figure 2 ijerph-18-01986-f002:**
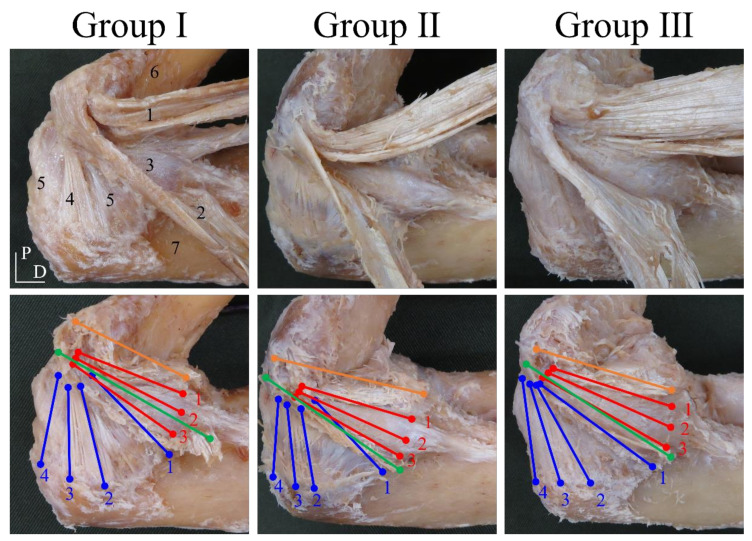
Classification of anterior bundle (AB), posterior bundle (PB), and common tendon (CT) and digitized parts (left side, medial view). Group I: the AB, PB, ACT, and PCT have independent forms. Group II: the AB, ACT, and PCT have independent forms, and the PB has an unclear form. Group III: The AB, PB, ACT, and PCT have unclear forms. 1: anterior common tendon (ACT). 2: posterior common tendon (PCT). 3: anterior bundle (AB). 4: posterior bundle (PB). 5: joint capsule. 6: humerus. 7: ulna. P: proximal. D: distal. Anterior common tendon (orange line): A line connecting the center of the distal end of the attachment and the proximal end of the tendon fibers located at the distal end. Posterior common tendon (green line): A line connecting the center of the distal end of the attachment and the proximal end of the tendon fibers located at the distal end. Anterior band of the AB (red line 1): Group I and Group II are the anterior edge of the AB, and Group III is the fiber bundle located at the posterior edge of the ACT. Central band of the AB (red line 2): A line connecting the midpoint between the origin of the anterior edge of the AB and the origin of the posterior edge of the AB, and the midpoint between the insertion of the anterior edge of the AB and the insertion of the posterior edge of the AB. Posterior band of the AB (red line 3): Group I and Group II are the posterior edge of the AB, and Group III is the fiber bundle located at the anterior edge of the PCT. Anterior band of the PB (blue line 1): Fiber bundle or joint capsule located at the posterior edge of the PCT. Anterior central band of the PB (blue line 2): Group I is the anterior edge of the PB, and Group II and Group III are the fiber bundles or the joint capsules located at about 2 mm anterior from the medial epicondyle tips of the humerus. Posterior central band of the PB (blue line 3): Group I is the posterior edge of the PB, and Group II and Group III are the fiber bundles or the joint capsules located at about 2 mm posterior from the medial epicondyle tips of the humerus. Posterior band of the PB (blue line 4): Fiber bundle or joint capsule located at the medial proximal end of the olecranon.

**Figure 3 ijerph-18-01986-f003:**
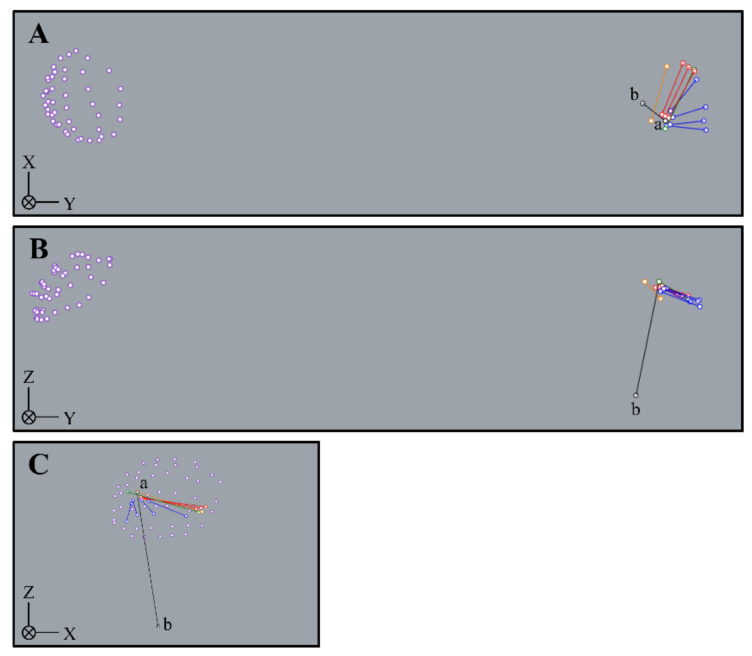
Three-dimensional construction of the AB, PB, ACT, and PCT (left side). (**A**): medial view. (**B**): vertical view. (**C**): distal view. a: medial epicondyle tip of the humerus. b: lateral epicondyle tip of the humerus. Purple point: head of the humerus. Orange line: anterior common tendon. Green line: posterior common tendon. Red line: anterior bundle. Blue line: posterior bundle.

**Figure 4 ijerph-18-01986-f004:**
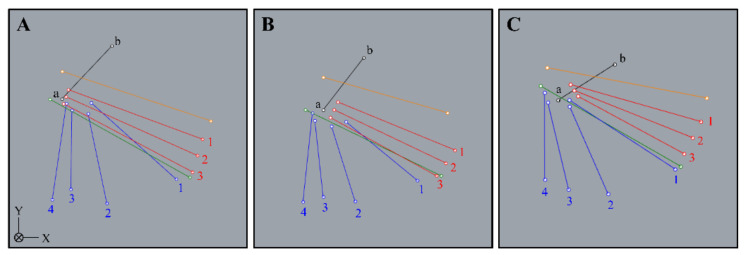
Three-dimensional construction of each group. (**A**): The AB, PB, ACT, and PCT are independent forms (Group I). (**B**): The AB, ACT, and PCT have independent forms, and the PB has an unclear form (Group II). (**C**): The AB, PB, ACT, and PCT have unclear forms (Group III). a: medial epicondyle tip of the humerus. b: lateral epicondyle tip of the humerus. Orange line: anterior common tendon. Green line: posterior common tendon. Red line 1: anterior band of the anterior bundle. Red line 2: central band of the anterior bundle. Red line 3: posterior band of the anterior bundle. Blue line 1: anterior band of the posterior bundle. Blue line 2: anterior central band of the posterior bundle. Blue line 3: posterior central band of the posterior bundle. Blue line 4: posterior band of the posterior bundle.

**Figure 5 ijerph-18-01986-f005:**
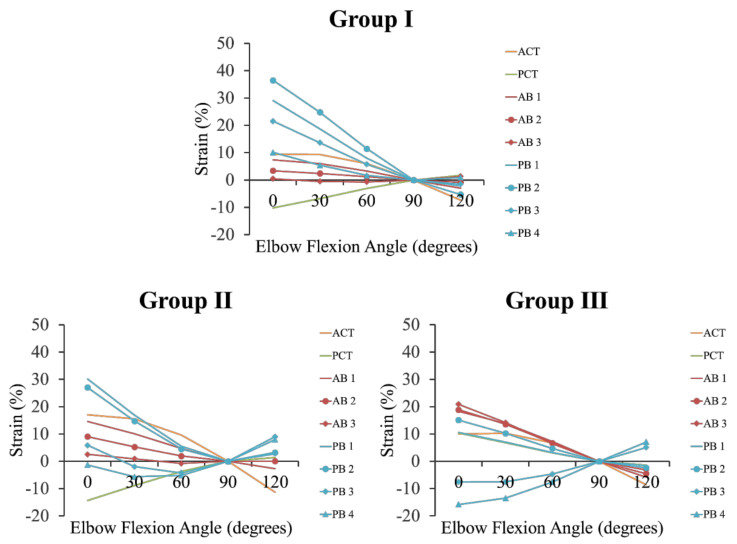
Changes in strain for each group during flexion and extension. Group I: The AB, PB, ACT, and PCT have independent forms. Group II: The AB, ACT and PCT have independent forms, and the PB has an unclear form. Group III: The AB, PB, and CT have unclear forms. ACT: anterior common tendon. PCT: posterior common tendon. AB 1: anterior band of the anterior bundle. AB 2: central band of the anterior bundle. AB 3: posterior band of the anterior bundle. PB 1: anterior band of the posterior bundle. PB 2: anterior central band of the posterior bundle. PB 3: posterior central band of the posterior bundle. PB 4: posterior band of the posterior bundle.

**Figure 6 ijerph-18-01986-f006:**
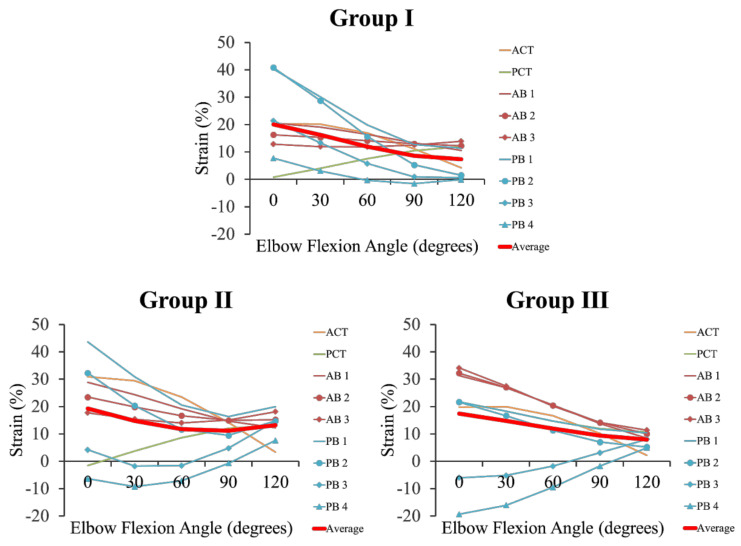
Changes in strain when elbow valgus (10°) is added at elbow joint flexion and extension for each group. Group I: The AB, PB, ACT, and PCT have independent forms. Group II: The AB, ACT, and PCT have independent forms, and the PB has an unclear form. Group III: The AB, PB, ACT, and PCT have unclear forms. ACT: anterior common tendon. PCT: posterior common tendon. AB 1: anterior band of the anterior bundle. AB 2: central band of the anterior bundle. AB 3: posterior band of the anterior bundle. PB 1: anterior band of the posterior bundle. PB 2: anterior central band of the posterior bundle. PB 3: posterior central band of the posterior bundle. PB 4: posterior band of the posterior bundle.

**Table 1 ijerph-18-01986-t001:** Changes in strain for each group during flexion and extension.

Group	0°	30°	60°	90°	120°
ACT					
Group I	9.4	9.3	6.0	0	−7.3
Group II	17.0	15.6	9.6	0	−11.3
Group III	10.0	10.3	6.7	0	−8.6
PCT					
Group I	−10.2	−6.8	−3.1	0	1.7
Group II	−14.3	−8.9	−3.7	0	1.4
Group III	10.3	6.8	3.1	0	−1.4
AB 1					
Group I	7.4	6.0	3.3	0	−3.0
Group II	14.6	10.1	4.7	0	−2.7
Group III	18.3	13.6	6.7	0	−4.5
AB 2					
Group I	3.4	2.4	1.2	0	−0.8
Group II	9.1	5.3	1.9	0	0.1
Group III	18.9	13.6	6.7	0	−4.5
AB 3					
Group I	0.5	−0.5	−0.7	0	1.4
Group II	2.6	0.9	−0.8	0	3.0
Group III	20.9	14.1	6.5	0	−3.2
PB 1					
Group I	29.1	18.7	8.0	0	−2.1
Group II	30.1	16.9	5.5	0	2.8
Group III	10.5	7.0	3.2	0	−1.6
PB 2					
Group I	36.5	18.7	8.0	0	−2.1
Group II	27.1	14.8	4.5	0	3.2
Group III	15.1	10.2	4.7	0	−2.4
PB 3					
Group I	21.5	13.7	5.7	0	−1.5
Group II	5.9	−1.9	−4.2	0	9.0
Group III	−7.6	−7.5	−4.7	0	5.1
PB 4					
Group I	10.1	5.4	1.7	0	1.0
Group II	−1.3	−5.7	−5.2	0	7.9
Group III	−15.8	−13.5	−7.6	0	7.0

Value: strain (%) at elbow flexion and extension. Group I: The AB, PB, ACT, and PCT have independent forms. Group II: The AB, ACT, and PCT have independent forms, and the PB has an unclear form. Group III: The AB, PB, ACT, and PCT have unclear forms. ACT: anterior common tendon. PCT: posterior common tendon. AB 1: anterior band of the anterior bundle. AB 2: central band of the anterior bundle. AB 3: posterior band of the anterior bundle. PB 1: anterior band of the posterior bundle. PB 2: anterior central band of the posterior bundle. PB 3: posterior central band of the posterior bundle. PB 4: posterior band of the posterior bundle.

**Table 2 ijerph-18-01986-t002:** Changes in strain when elbow valgus (10°) is added at elbow joint flexion and extension for each Group.

Group	0°	30°	60°	90°	120°
ACT					
Group I	20.3	20.2	16.9	11.2	4.3
Group II	30.9	29.4	23.5	14.2	3.3
Group III	19.8	20.0	16.6	10.2	2.2
PCT					
Group I	0.8	4.0	7.6	10.5	12.0
Group II	−1.5	3.7	8.7	12.1	13.2
Group III	21.7	18.3	14.7	11.9	10.7
AB 1					
Group I	20.5	19.1	16.5	13.4	10.5
Group II	28.9	24.4	19.2	14.7	12.3
Group III	31.3	26.9	20.6	13.8	8.4
AB 2					
Group I	16.3	15.3	14.1	13.0	12.3
Group II	17.8	15.4	14.0	15.0	18.1
Group III	34.1	27.5	20.2	14.2	11.4
AB 3					
Group I	23.5	19.8	14.1	13.0	12.3
Group II	17.8	15.4	16.7	15.0	15.2
Group III	34.1	27.5	20.2	14.2	11.4
PB 1					
Group I	21.4	13.4	5.8	0.9	0.6
Group II	43.7	30.9	20.5	16.3	20.0
Group III	21.7	18.3	14.7	11.7	10.4
PB 2					
Group I	40.8	28.8	15.7	5.3	1.5
Group II	32.3	20.3	11.6	9.5	14.8
Group III	21.7	16.7	11.3	7.0	5.3
PB 3					
Group I	21.4	13.4	5.8	0.9	0.6
Group II	4.2	−1.8	−1.5	4.8	14.7
Group III	−6.0	−5.2	−1.7	3.1	8.1
PB 4					
Group I	7.7	3.1	−0.3	−1.5	−0.1
Group II	−6.3	−9.3	−7.1	−0.7	7.6
Group III	−19.4	−16.1	−9.5	−1.8	5.0
Average					
Group I	20.1	16.2	12.0	8.7	7.4
Group II	19.3	14.8	11.7	11.2	13.3
Group III	17.5	14.8	11.9	9.4	7.9

Value: strain (%) at elbow valgus. Group I: The AB, PB, ACT, and PCT have independent forms. Group II: The AB, ACT, and PCT have independent forms, and the PB has an unclear form. Group III: The AB, PB, ACT, and PCT have unclear forms. ACT: anterior common tendon. PCT: posterior common tendon. AB 1: anterior band of the anterior bundle. AB 2: central band of the anterior bundle. AB 3: posterior band of the anterior bundle. PB 1: anterior band of the posterior bundle. PB 2: anterior central band of the posterior bundle. PB 3: posterior central band of the posterior bundle. PB 4: posterior band of the posterior bundle.

## Data Availability

The datasets used and/or analyzed during the current study are available from the corresponding author on reasonable request.

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
