# Peer review of "The Effects of Differences in the Morphologies of the Ulnar Collateral Ligament and Common Tendon of the Flexor-Pronator Muscles on Elbow Valgus Braking Function: A Simulation Study"

_ijerph, 2021, doi:10.3390/ijerph18041986_

Round 1

Reviewer 1 Report

The study was to construct 3-D models of the medial elbow ligaments and muscular structures on elbow valgus braking function by simulation. The article is basically well-written and clear description. There are only a few questions to the authors.

1.Do the authors provide any clinical significance based on the current results ?

2.The previous studies have reported the role of elbow valgus stabilizer from           ulnar collateral ligament and the flexor-pronator muscles, what is the updated     opinion from the current study ?  

Reviewer 2 Report

This study evaluated the function of the medial collateral ligament on the valgus. However, the number of specimens is completely insufficient to classify ligament structures.

Furthermore, formalin-fixed specimens are not suitable for evaluation as a biomechanics study of soft tissues. Fresh-frozen cadavers should be used for this study, and it is recommended that power analysis be performed appropriately to determine the number of specimens.

Line 63 It is unsuitable to perform a biomechanics study on soft tissues in an elbow joint fixed with formalin. The effect of formalin fixation on the assessment of strain and anatomical structures needs to be reconsidered.

Line 70  The number of cases in each group classified by ligament structure was only 1. Each specimen may not necessarily represent the result of that ligament structure. It is inevitable that the specimen will be subject to other specimen-specific effects, such as joint geometry, cartilage loss, and osteophyte formation, etc. The number of cases needs to be reconsidered.

Line 132  How did you measure the angle of the elbow joint? Please indicate how you measured the angle of flexion and valgus.

Line132  How did you apply the 10° valgus and maintain that position? Did you experience any viscoelasticity or other effects from the sustained valgus?

Reviewer 3 Report

The authors did a great effort of notable methodological quality covering an interesting and slightly covered topic. The primary aim of the study was to investigate and better clarify the morphological differences of AB, PB,  and CT by 3D models constructs from cadaver models and evaluate the possible implications on elbow valgus braking function by simulation models. As mentioned, the idea can be considered objectively valid and novel with potential significant theoretical and practical implications. As described by the authors one population that can take advantages from insights on this field are throwers and athlete’s staff aiming to prevent injuries and improve throwing technique.

As already mentioned, the novelty of the study, the methodological quality and possible theoretical and practical implications arising from such investigation represent the strengths of this investigation. However, a suggestion could be to further remark in the discussions and conclusion section the possible practical implications of these findings. The authors stated in one of the first sentences that “Ulnar collateral ligament injury is among the frequent sports injuries of throwing athletes” however there’re no or few refence to this in the discussion nor in the conclusion.

Line 50

“…while others have stated that…” instead of “…and others have stated that…”

Line 52

“…controversial…” instead of “…inconsistent…” (to avoid repetition of line 46 and to use a more appropriate term since the citated studies reported opposite statements)

Line 54

“…an independent and unclear form…”

Line 248 and 249

These sentences could be structured in a better way.

Round 2

Reviewer 2 Report

In view of the fact that the specimens for the Biomechanical Study are fixed in formalin and the number of cases in each group is only 1, it is difficult to believe that the correct conclusions can be drawn. Therefore, I regret to say that the rejection of this study is appropriate.
